# Influence of Mycorrhiza on C:N:P Stoichiometry in Senesced Leaves

**DOI:** 10.3390/jof9050588

**Published:** 2023-05-18

**Authors:** Shan-Wei Wu, Zhao-Yong Shi, Ming Huang, Shuang Yang, Wen-Ya Yang, You-Jun Li

**Affiliations:** 1College of Agriculture, Henan University of Science and Technology, Luoyang 471023, China; 2Luoyang Key Laboratory of Symbiotic Microorganism and Green Development, Luoyang 471023, China; 3Henan Engineering Research Center of Human Settlements, Luoyang 471023, China

**Keywords:** biogeochemical cycling, carbon, nitrogen, phosphorus, senesced leaves, mycorrhizal strategy, temperature and precipitation

## Abstract

Senesced leaves play a vital role in nutrient cycles in the terrestrial ecosystem. The carbon (C), nitrogen (N) and phosphorus (P) stoichiometries in senesced leaves have been reported, which are influenced by biotic and abiotic factors, such as climate variables and plant functional groups. It is well known that mycorrhizal types are one of the most important functional characteristics of plants that affect leaf C:N:P stoichiometry. While green leaves’ traits have been widely reported based on the different mycorrhiza types, the senesced leaves’ C:N:P stoichiometries among mycorrhizal types are rarely investigated. Here, the patterns in senesced leaves’ C:N:P stoichiometry among plants associated with arbuscular mycorrhizal (AM), ectomycorrhizal (ECM), or AM + ECM fungi were explored. Overall, the senesced leaves’ C, with 446.8 mg/g in AM plants, was significantly lower than that in AM + ECM and ECM species, being 493.1 and 501.4 mg/g, respectively, which was mainly caused by boreal biomes. The 8.9 mg/g senesced leaves’ N in ECM plants was significantly lower than in AM (10.4 mg/g) or AM + ECM taxa (10.9 mg/g). Meanwhile, the senesced leaves’ P presented no difference in plant associations with AM, AM + ECM and ECM. The senesced leaves’ C and N presented contrary trends with the changes in mean annual temperature (MAT) and mean annual precipitation (MAP) in ECM or AM + ECM plants. The differences in senesced leaves’ C and N may be more easily influenced by the plant mycorrhizal types, but not P and stoichiometric ratios of C, N and P. Our results suggest that senesced leaves’ C:N:P stoichiometries depend on mycorrhizal types, which supports the hypothesis that mycorrhizal type is linked to the evolution of carbon–nutrient cycle interactions in the ecosystem.

## 1. Introduction

Senesced leaves play an important role in plant nutrient resorption, which impacts nutrient cycling, plant productivity and vegetation community structure [1,2,3]. Leaf nutrients are resorbed from senesced leaves to keep more nutrients in plants, which is an important strategy for plants to conserve nutrients [4]. More than half of the nutrients in leaves can be resorbed during senescence [1], including the mobile proteins, carbohydrates, and other nutrients [5,6]. Therefore, plant nutrient resorptions effectively reduce the dependence on soil nutrients and improve productivity when the soil has low nutrient availability [7,8]. In particular, N and P resorption from senesced leaves mainly translocated into reproductive growth, increasing gain yield [9,10]. A global study reported that nitrogen (N) and phosphorous (P) in senesced leaves were significantly related to latitude and climate factors [11]. Previous studies have shown that climate, plant species and soil microbial composition are the main factors affecting the nutrient levels of senesced leaves [12,13]. Furthermore, the nutrients in senesced leaves also influence soil nutrient availability and carbon sequestration, with significant feedback on climate [14,15]. Thus, senesced leaves’ traits have aroused increasing attention, particularly in regard to the correlation between mycorrhizal strategy and plant stoichiometry [16,17].

Carbon (C), N and P play a vital role in maintaining and balancing plant nutrients [18,19,20,21]. The C:N:P stoichiometry is an important indicator of biochemical processes, such as community organization, nutrient limitation and decomposition [22,23,24]. In particular, leaf stoichiometry is very sensitive to nutrient availability [25,26]. Numerous studies have found that litter decomposition, physiological processes and other plant functionalities are influenced by leaf C:N:P stoichiometry in the ecosystem [27,28,29]. Yan et al. [30] have also concluded that there is a strong correlation between leaf stoichiometry and water use efficiency in plants. Meanwhile, leaf C:N:P stoichiometry is significantly affected by many factors, including climate [31,32], external nutrient availability [33,34], arbuscular mycorrhiza [16,17] and environmental conditions [35,36]. However, the relationship between senesced leaves’ stoichiometry and plant mycorrhizal association remains unknown.

Mycorrhizal fungi are vital members of the plant microbiota that can form a symbiotic relationship with the root of over 92% plants species on the earth [37,38]. A large number of studies have shown that mycorrhizal fungi could influence ecosystem functions, such as soil microbial community composition, the plant and soil nutrient cycle, and soil carbon stabilization [39,40,41,42,43]. The impact of mycorrhizal fungi on nutrient absorption may depend on the plant species, for example, forbs vs. grasses [44]. Hoeksema et al. [45] also found that forb and woody plants had a more positive response to mycorrhizal inoculation than grasses. The woody plants had higher mycorrhizal inoculation densities than herbs because there are more passage cells and penetration points in woody plants [46]. Moreover, there were different responses to plant growth in different mycorrhizal types [47]. On the one hand, mycorrhizal types affect the nutrient absorption and resistance of host plants through altering the soil microbial community [48]. On the other hand, mycorrhizal types also influence plant coexistence which is usually linked to soil nutrient resources, especially N and P [49]. Mycorrhizal symbiosis can be divided into many types according to its morphology and the identity of its partners. Arbuscular mycorrhizal (AM) and ectomycorrhizal (ECM) are the two most common mycorrhizal types [50]. AM plants were the most abundant plant type in all biomes because AM fungi can symbiose with almost 79% of terrestrial plant species [51]. By contrast, ECM plants have lower plant diversity than AM, which are dominant in a high latitude ecosystem [51,52]. Previous studies have shown that the differences between AM and ECM plants reflect biogeochemical changes in ecosystems [52]. AM plants mainly rely on inorganic nitrogen with rapid nutrient cycling [52,53,54], while ECM plants are able to use organic nitrogen with slow nitrogen cycling [55]. The differences in nutrient acquisition strategies lead to variation in nutrient cycling between AM and ECM associations [56,57]. There were many studies that explored why ECM plants have more conservative nutrient cycles and associated lower nutrient concentrations in green leaves than AM plants [55,58,59,60]. More and more studies found that the leaf nutrient resorption efficiency decreased with increasing nutrient concentrations [1,5,61]. Therefore, the ECM plants have higher leaf nutrient resorption and slower litter decomposition rates than AM plants. A few studies have focused on the leaf traits of plants with different mycorrhizal types, but no definitive conclusion can be drawn on a global scale. Shi et al. [16] draw a conclusion that mycorrhiza types are closely related to the leaf economic spectrum. Averill et al. [37] found that ECM plants have more conservative nutrient cycles and lower leaf nutrient contents than AM. According to reports, the leaf litters of AM plants decayed faster than the litters of ECM plants in temperate forests, which was mainly caused by litter N and phylogeny [62]. However, the relationship between mycorrhizal types and senesced leaves’ C:N:P stoichiometry remains unknown.

Understanding whether mycorrhizal statues can be used as an important indicator for plant nutrient resorption is crucial for improving plant production by maintaining plant fitness with regard to environmental changes [63]. Here, we combined a dataset which included C, N and P content and their stoichiometric ratio in senesced leaves in forbs, grasses and woody plants, with information on mycorrhizal associations, leaf habits, leaf shapes and biomes. Regarding the effects of mycorrhizal types on leaf nutrients [16,17], the resorption of nutrients might be used to evaluate the potential for sustainable plant production based on changes in the contents of senesced leaves. The purpose of this study was to reveal the effect of mycorrhizal types on the distribution in senesced leaves of C, N and P and their stoichiometric ratios and to understand the plant resorption strategy to cope with the environment. We hypothesized that (1) mycorrhizal types alter plant senesced leaves C, N and P content and their stoichiometric ratio; (2) the effect of mycorrhizal types on plant senesced leaves C, N and P and their stoichiometric ratio is closely related to plant functional groups and climate variables.

## 2. Materials and Methods

### 2.1. Data Collection

In this study, the plant senesced leaves’ C:N:P stoichiometry data were obtained from the global database established by Yuan and Chen [64]. The 1253 observations were obtained from two leaf habits (deciduous vs. evergreen), two leaf shapes (broadleaf vs. conifer leaf shape) and three biomes (boreal, temperate and tropical). A new database containing senesced leaves’ C:N:P stoichiometry and associated traits was established by rearranging the database from Yuan and Chen [64].

### 2.2. Mycorrhizal Classification

The mycorrhizal types of all the plant species were classified according to the method employed by Shi et al. [16] and Yang et al. [17]. The mycorrhizal types were ascertained according to the published literature, mainly including Wang and Qiu [65], Akhmetzhanova et al. [66], Hempel et al. [67] and Soudzilovskaia et al. [68]. According to all the data we collected, the species forming AM or ECM account for 91.9%. In addition, AM and ECM fungi are two domain functional types inoculated with most plants on Earth [57]. Therefore, we classified the mycorrhizal type as AM, AM + ECM or ECM. Our data extend to all species to avoid the effect of plant species-specific, ecological and evolutionary strategies, from the same species in different sites to the same species in the same site. On this basis, a database was established by combining the identified mycorrhizal types of plant species with data on senesced leaves C, N and P, and their stoichiometric ratios and associated traits (Supplementary File). Additionally, the database contains 895 observations of mycorrhizal association, deciduous vs. evergreen leaf habit, broadleaf vs. conifer leaf shape, biomes, and senesced leaves’ element contents across 397 plant species (Figure 1). In the new senesced leaves dataset, there were 244 (61.5%), 45 (11.3%) and 108 (27.2%) plant-formed host-specific associations with AM, AM + ECM, or AM + ECM, respectively. Plant species were subdivided into two subgroups based on biomes and plant functional types. The biome was further divided into three sub-subgroups, i.e., boreal, temperate and tropical. The plant functional types were further divided into two sub-subgroups based on leaf shape (i.e., broadleaf and conifer) and leaf habit (i.e., deciduous and evergreen) (Appendix A).

### 2.3. Data Analysis

To confirm the class of all the species in this study, the phylogenetic tree was constructed within the “phytools” package in R v.4.2.1 (R Core Team, 2022) [37,69]. In order to compare the contents of senesced leaves C, N and P and their stoichiometric ratios in the plant associated with different mycorrhizal types, permutation tests were performed with 9999 permutations. We examined the effects of plant functional types as classified by leaf shape (broadleaf versus conifer) and leaf habit (deciduous versus evergreen) on senesced leaves’ nutrients using permutation tests. The effects of mycorrhizal types, leaf shape and leaf habit on the nutrient contents of senesced leaves and their stoichiometric ratios were examined by linear mixed effect models. Each senesced leaf’s trait was treated as a response variable. Leaf shape, mycorrhizal type and their interaction were fixed effects. Leaf habit was considered as a random effect. The analyses of the permutation test and linear mixed effect models were conducted in R v.4.2.1. Based on Shi et al. [16] and Wright et al. [70], the C, N and P contents were log10-transformed in order to improve the normality of distributions and homogeneity assumptions, but they were shown as untransformed values for easy understanding. Then, we tested if the responses of senesced leaves’ nutrients in regard to MAT and MAP differed among mycorrhizal types via linear regression analysis. The standardized regression slopes were calculated to determine the relationship between senesced leaves’ nutrients and MAT/MAP. The linear regression analysis was performed using SPSS 22.0. Data are presented as mean ± standard deviation. All the figures were generated using Excel 2016.

## 3. Results

### 3.1. The Contents of Senesced Leaves’ Carbon, Nitrogen and Phosphorus and Their Stoichiometric Ratios among Plant Associations with Different Mycorrhizal Types

Mycorrhizal types had significant effects on senesced leaves’ C, N and N:P ratio, but not on P, C: N and N:P ratios (Figure 2). The C content in senesced leaves was 446.8 mg/g, 493.1 mg/g and 501.4 mg/g in association with AM, AM + ECM and ECM (Figure 2A), respectively. The senesced leaves N in AM, AM + ECM and ECM plants were 10.4 mg/g, 10.9 mg/g and 8.9 mg/g, respectively (Figure 2B). ECM plants had significantly higher C and lower N in senesced leaves than AM-associated plants. Similarly, the N:P ratio in ECM plants was lower than in AM and AM + ECM plants. The N:P ratios in AM, AM + ECM and ECM plants were 20.5, 20.1 and 16.1, respectively (Figure 2F). Meanwhile, the senesced leaves’ P, C:N and C:P were not remarkably different in AM, AM + ECM and ECM plants (Figure 2C–E).

For different climate zones, the effect of mycorrhizal type on senesced leaves’ C:N:P stoichiometry presented different trends (Table 1). In the boreal, the ECM (512.3 mg/g) and AM + ECM (501.4 mg/g) plants had significantly higher C in the senesced leaves than AM (460.1 mg/g). Meanwhile, the C content in senesced leaves did not exhibit significant differences in AM, AM + ECM and ECM plants in the temperate and tropical zones. The senesced leaves’ N:P ratio in AM, AM + ECM and ECM plants presented a similar trend to that of the senesced leaves’ C content. Among the three mycorrhizal types, AM + ECM had the highest N (11.7 mg/g) in senesced leaves, whereas ECM plants had the lowest value (7.3 mg/g) in the boreal zone. There was no significant variation in senesced leaves’ N in AM, AM + ECM and ECM plants in temperate and tropical zones. The tropical C:N ratio was 39.7, 53.0 and 35.3 in AM, AM + ECM and ECM plants, respectively, and the C:N ratio in AM + ECM plants was significantly higher than in AM and ECM species. Boreal and temperate C:N ratio had no remarkable difference to that in AM, AM + ECM and ECM plants. In addition, there was no significant variation in senesced leaves’ P and C:P ratios in association with the three mycorrhizal types under three climate zones.

Plant functional type (leaf shape and leaf habit) also affected C:N:P stoichiometry in senesced leaves among three mycorrhizal types (Table 2). For deciduous species, only senesced leaves’ C presented a remarkable difference in plants associated with different mycorrhizal types. However, the ECM plants had lower N, N:P ratio and higher C in their senesced leaves than the other two mycorrhizal types in evergreen species. The N, P, N:P and C:N in senesced leaves showed no significance in AM, AM + ECM and ECM plants for broadleaf species, except for in the C content and C:P ratio. Meanwhile, conifer species only had significantly lower N content in ECM plants than it in AM and AM + ECM plants.

Linear mixed effect model analyses confirmed that the senesced leaves’ C, N and N:P ratios varied with mycorrhizal types, whereas only N:P ratios varied with the leaf shape. The significant interactions between mycorrhizal types and leaf shapes were only observed in senesced leaves’ N (Table 3).

### 3.2. The Regulation of Senesced Leaves’ Carbon, Nitrogen, Phosphorus and Their Stoichiometric Ratios by MAT and MAP among Plant Associations with Different Mycorrhizal Types

The relationship between senesced leaves’ C:N:P stoichiometry and MAT varied among three mycorrhizal types. Overall, the C content in senesced leaves decreased with increasing MAT (Figure 3). The slopes of senesced leaves’ C against MAT in AM, AM + ECM and ECM plants were −4.97 × 10^−3^, −4.1 × 10^−3^ and −1.51 × 10^−3^, respectively. The slope changed markedly in AM and ECM plants, and the same was true in AM and AM + ECM plants (Table 4). While the senesced leaves’ N increased with increasing MAT, its slope changed significantly among AM, AM + ECM and ECM plants. Similarly, the N:P ratio increased with MAT and the slope changed markedly among the three mycorrhizal types (Table 4). The senesced C:N:P stoichiometry differed in response to mycorrhizal types with MAP (Figure 4). Overall, the senesced leaves’ N and N:P ratio increased with increasing MAP in the three mycorrhizal types. The slopes of the senesced leaves’ N against MAP in AM, AM + ECM, and ECM plants were 2.39 × 10^−2^, 1.32 × 10^−2^ and 4.20 × 10^−2^, respectively, showing the significant difference in the slopes of different mycorrhizal types (Table 5). Similarly, the slope of senesced leaves’ N:P against MAP changed markedly in ECM and AM + ECM plants, and the same was true in AM and ECM plants.

## 4. Discussion

The objective of this study is to explore the senesced leaves’ C, N and P contents and their stoichiometric ratios among AM, AM + ECM and ECM plants. Our results showed that mycorrhizal types affected senesced leaves’ C, N and N:P ratio. According to Averill et al.’s research [37], the N content in senesced leaves also presented a significant difference between AM and ECM plants. These similar results indicated that patterns of nutrient cycles, which are related to mycorrhizal types, may be ecosystem-specific [71]. This study will help us to understand the mechanisms for plant nutrient strategies in plant associations with mycorrhizal fungi, and provide a prediction of the sensitivity of ecosystems to environmental changes.

Previous studies on the distribution of leaf C, N and P contents in different plant function types looked at shrubs [17,31], forests [72], grasslands [73] and terrestrial plants [74]. In particular, numerous studies focused on the distribution of senesced leaves’ N and P [37,64], which were related to climate, nutrient type and leaf habit. Recently, there have been many studies on green leaves’ C, N and P contents and their stoichiometric ratios in different mycorrhizal types and statuses. Shi et al. [16] pointed out that green leaves’ traits were significantly different in AM and non-AM plants at the global scale. Yang et al. [17] also drew a conclusion that the green leaves’ C, N and P contents were significantly different among AM, AM + ECM and ECM plants in northern China. Here, we first evaluated the senesced leaves’ C, N and P contents and their stoichiometric ratios among different mycorrhizal types and found significant differences in C and N contents in AM, AM + ECM and ECM plants.

Compared to AM plants, ECM and AM + ECM plants had significantly higher C content in senesced leaves, indicating that ECM and AM + ECM plants were better in promoting the C absorption from senesced leaves than AM plants. Yang et al. [17] determined that AM + ECM had a higher C content in shrub leaves than AM and ECM plants. Generally, the plants’ associations with ECM exhibit high nutrient resorption from soil compared with AMs [56]. Therefore, the ECM plants may promote plant growth and finally benefit from the accumulation of C in the plant. The N content in senesced leaves was the lowest in ECM, which was in contrast with the results of previous studies [75,76], but it was similar to Averill et al.’s [37] research. The reason may be attributed to ECM plants as they degrade and absorb the inorganic nitrogen in the soil, while AM plants depend on the inorganic nitrogen resources [37]. Moreover, ECM could absorb N from organic matter, while AMF promotes the microbial decomposition of organic matter to obtain mineral N [77]. Therefore, the ecosystem domain in ECM had greater soil carbon storage than the ecosystem domain in AM [78]. Further, N content in AM + ECM plants was the highest among AM, AM + ECM and ECM plants, indicating that the co-existence of AM and ECM could enhance the N absorption by plants. Moreover, the variation in senesced leaves’ P among the three mycorrhizal types was not significant, which was due to the low numbers of observation and more plant species in our database. Furthermore, there was no significant difference between C:N and C:P in AM, AM + ECM and ECM plants, which may be caused by the characteristics of plants. Yuan and Chen [25] also found that the C:N and C:P ratios in plants were not influenced by global climate change, such as increasing concentrations of CO_2_, rainfall, warming and drought. On the one hand, plant species are the main reason for the alterations in C:N and C:P ratios [79]; for example, C4 plants are considered more conservative than C3 plants in absorbing nutrients from the soil [80,81]. On the other hand, environmental factors had no potential to alter the balance of plant C and N in the ecosystem [32,82]. The senesced leaves’ N:P in plants with ECM was significantly lower than in AM and AM + ECM plants (Figure 2F), which was mainly caused by N and P content, as seen in Figure 2B,C. Previous studies also indicated that plant N:P ratios in natural environments had a strong response to environmental change [21,26,83,84,85]. These findings imply that the mycorrhizal symbiosis could represent carbon–nutrient cycle interactions in the ecosystem because the plant associations with different mycorrhizal types had different nutrient acquisition strategies [86,87].

For different biomes, the effects of mycorrhizal type on senesced leaves’ C, N and P contents and their stoichiometric ratios were also different (Table 1). The ECM plants had higher C content and lower N content in senesced leaves than the other mycorrhizal types, which is consistent with Averill et al. [37]. The reasons can be explained by plant nutrient strategies. On the one hand, ECM plants use nitrogen more conservatively than AM plants in boreal zones [37]. On the other hand, the plants with greater nutrient resorption (such as AM plants) were known to allocate more carbon belowground and thus contribute to reducing the accumulation of carbon in the leaf [52]. It can be inferred that the nutrient use strategies of ECM plants in boreal zones may be more conserved than those of AM plants [76]. Furthermore, AM plants could promote greater soil carbon loss to improve N uptake from the soil compared to ECM because they enhance organic matter decomposition [63]. Moreover, the senesced leaves’ N in AM + ECM plants has been shown to be the highest out of the three different mycorrhizal types. This suggests that the coexistence of AM and ECM may have some synergetic effects to improve the absorption of nitrogen from soil in plants. Similarly, the ECM plants had significantly lower N:P than plants with AM, as there were lower N resorption efficiencies in senesced leaves in ECM plants [88]. Numerous studies have proven that there is no variability in senesced leaves’ C and N among different mycorrhizal types in temperate and tropical zones [37,76,89], which is in contrast with our results. Chuyong et al. [89] pointed out that nitrogen resorption was nearly two times lower in ECM plants than AM plants in tropical zones. First of all, warm and wet climates ensure rapid litter decay [90]; therefore, nutrient resorption may be less vital for plant nutrient acquisition in temperate and tropical zones. The second explanation is probably attributed to our database, which includes less species than other studies [37,76,89]. These results indicated that plant associations with different mycorrhizal types had remarkable variability in nutrient acquirement, which also affects soil C [91]. In addition, plant C:N:P stoichiometries were mainly related to the plant species, which were caused by the genetic characteristics of the plant [17,74,79]. The senesced leaves’ C:N shows the differences between AM, AM + ECM and ECM plants. It is possible that the significant difference reflects the low number of observations for C:N in the tropical zone.

We have observed that the senesced leaves’ C, N and P content and their stoichiometric ratios had variations among AM, AM + ECM and ECM plants in different plant functional types (Table 2), which is similar to the results of Yang et al. [17] and Zhang et al. [76]. The reason for this can be explained by the plants inherently. AM plants inherently have a higher growth rate than plants with other mycorrhizal types, which require high N and N:P to maintain high rates of physiological processes [59]. Furthermore, AM plants had higher N resorption to allocate more carbon belowground and thus contribute to reducing the accumulation of carbon in the leaf. Regarding the leaf habit, evergreen species generally dominate in a nutrient poor environment, and have low nutrient contents and a slow growth rate. Therefore, the effects of nutrient resorption in AM evergreen species were significantly higher than in other mycorrhizal types. Similarly, in previous studies, the differentiation in leaf nutrient content among different mycorrhizal types was mostly attributed to deciduous and evergreen habits [76,79]. Beyond our expectation, there was no significant overall difference in senesced leaves’ P content among AM, AM + ECM and ECM plants. The main reason was the low number of observations for senesced leaves’ P in our database. Furthermore, many studies also have shown that the contents of C and N in leaves vary with the mycorrhizal type [16,37,71], which is consistent with our findings (Table 3). This result shows that mycorrhizal types have greater control over senesced leaves’ C and N contents than leaf shape. This may be caused by the different effect of mycorrhiza on plant resorption.

The climate (precipitation and temperature) can explain the variability in global nutrient resorption. The changes in temperature and precipitation can directly influence soil biogeochemical processes and vegetative composition, which affect the plant nutrients [92,93,94]. The climate, especially temperature, was the main factor that shaped the distribution of AM and ECM host plants [95]. Therefore, the variations in senesced leaves’ C and N contents in AM, AM + ECM and ECM plants were mainly related to MAT. Numerous studies also concluded that the N content in senesced leaves increased with increasing MAT [64,96], which is proven in our research. For example, Norby et al. [96] found that the senesced leaves’ N was related to temperature. Similarly, senesced leaves’ N, which increased with increasing MAT, showed the same pattern in green leaves [74,97]. However, the senesced leaves’ C was negatively related to MAT, especially for AM plants, which is the opposite of that for senesced leaves’ N. The reason is mainly related to the fact that the more nitrogen that is resorbed from leaves, the more carbon that is distributed to the underground of the plant [52]. MAP is also related to senesced leaves’ nutrients in AM, AM + ECM and ECM plants (Figure 4). Because plant nitrogen uptake and transport are closely related to soil moisture [98,99], increasing MAP was likely to increase green leaves’ N and eventually increase senesced leaves’ N. Therefore, different mycorrhizal types have different influences on plant adaptions to climate change. These results could imply that plant associations with different mycorrhizal types contributed differently to nutrient strategies in the ecosystem [100]. Considering the distinct patterns of climate factors, and senesced leaves’ nutrients among plant associations with different mycorrhizal types, our findings suggested that mycorrhizal strategies can explain the pathways of plant nutrient acquisition among species under climate change conditions.

## 5. Conclusions

The senesced leaves’ C and N contents differed in plants associated with AM, AM + ECM and ECM at the global scale, which were greatly linked to biomes and plant functional types. The P content in senesced leaves does not vary with mycorrhizal type. For the C:N:P ratios of senesced leaves, only N:P displayed large variation with the differences in mycorrhizal types, but not C:N and C:P. The significant effects of mycorrhizal types on senesced leaves’ C, N and N:P were attributed to species in the boreal zone not to those in tropical and temperate zones. For different plant functional types, mycorrhizal types influenced senesced leaves’ C and N contents and N:P ratios in evergreen plants. These results investigate the effects of mycorrhizal types on senesced leaves’ C:N:P stoichiometries and provide insights into the mechanisms of plant nutrient reacquisition strategies. Overall, our findings were able to accurately predict that the responses of plant C:N:P stoichiometries in senesced leaves to climate changes differed among mycorrhizal types.

## Figures and Tables

**Figure 1 jof-09-00588-f001:**
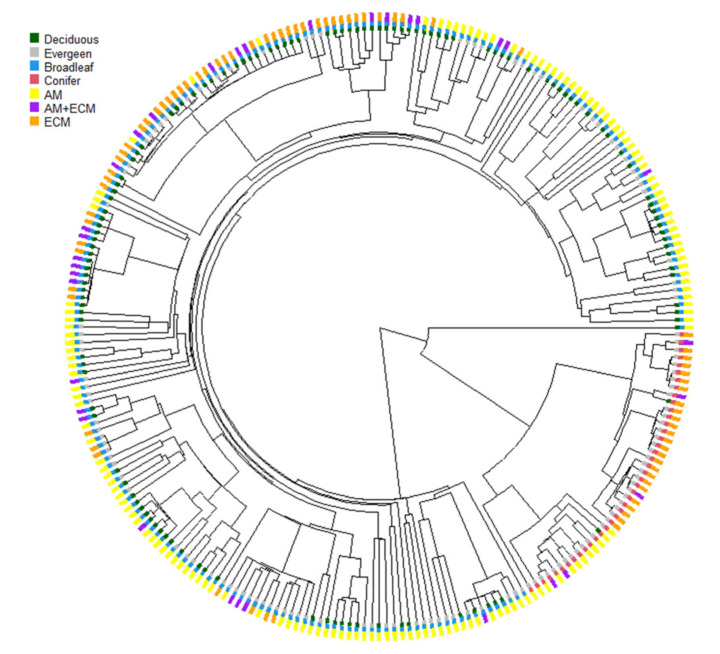
Phylogeny tree of all plant species used in this study. Plants in phylogenetic tree are shown with their mycorrhizal association (arbuscular mycorrhizal (AM), ectomycorrhizal (ECM), or AM + ECM), leaf shape (broadleaf and conifer), and leaf habit (deciduous and evergreen).

**Figure 2 jof-09-00588-f002:**
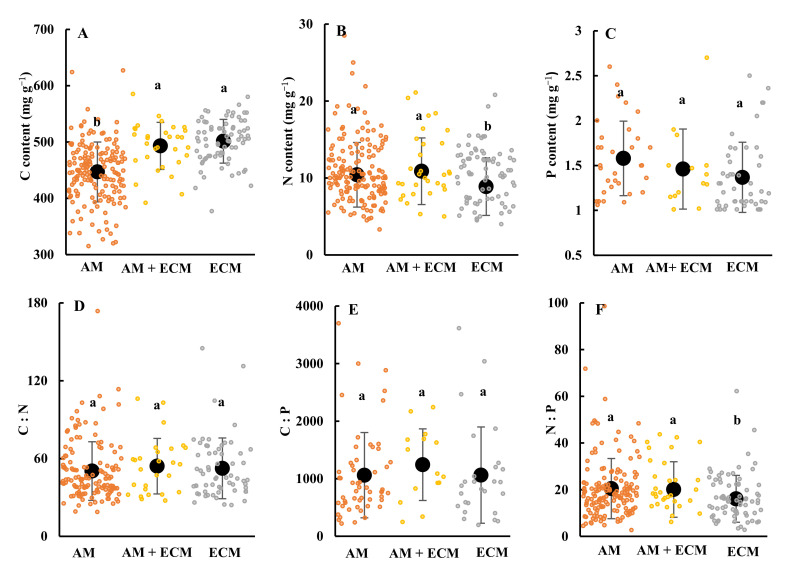
Senesced leaves’ carbon (C) (Figure **A**), nitrogen (N) (Figure **B**), phosphorus (P) (Figure **C**), C:N (Figure **D**), C:P (Figure **E**), N:P (Figure **F**) in plant association with AM, AM + ECM, and ECM. The orange solid cycles represent the content and stoichiometric ratios of elements in AM; the yellow solid cycles represent the content and stoichiometric ratios of elements in AM + ECM; the gray solid cycle represents the content and stoichiometric ratios of elements in ECM. The black point means the average value of each parameter. Line bars show the standard deviation. Letters above the line bars show significant difference among AM, AM + ECM and ECM mycorrhizal types at *p* < 0.01.

**Figure 3 jof-09-00588-f003:**
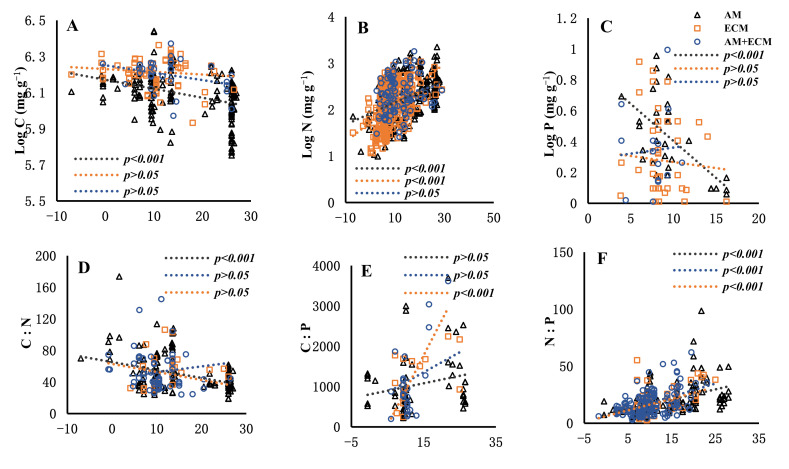
Senesced leaves’ carbon (C) (Figure **A**), nitrogen (N) (Figure **B**), phosphorus (P) (Figure **C**) C:N (Figure **D**), C:P (Figure **E**), N:P (Figure **F**) in relation to mean annual temperature (MAT, °C) in plant associations with AM, AM + ECM and ECM. The gray triangle and gray line mean AM, the blue circle and blue line mean AM + ECM, and the orange rectangle and orange line mean ECM. The *p* values are shown in each panel.

**Figure 4 jof-09-00588-f004:**
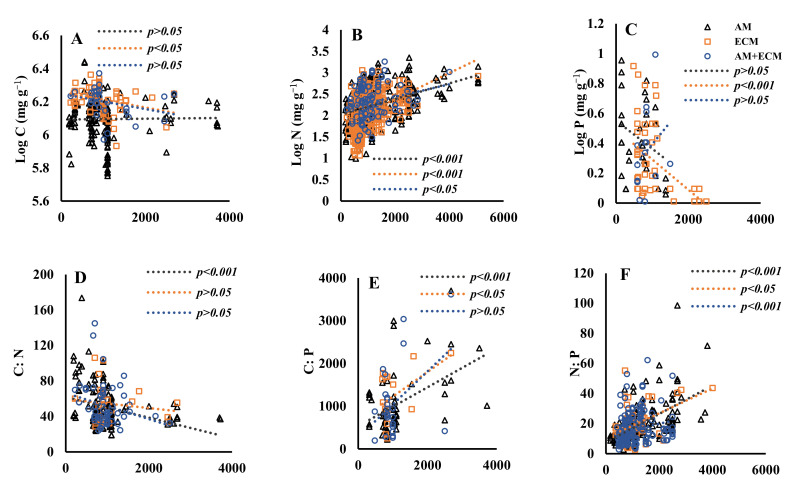
Senesced leaves’ carbon (C) (Figure **A**), nitrogen (N) (Figure **B**), phosphorus (P) (Figure **C**) C:N (Figure **D**), C:P (Figure **E**), N:P (Figure **F**) in relation to mean annual precipitation (MAP, mm) in plant associations with AM, AM + ECM and ECM. The gray triangle and gray line mean AM, the blue circle and blue line mean AM + ECM, and the orange rectangle and orange line mean ECM. The *p* values are shown in each panel.

**Table 1 jof-09-00588-t001:** The senesced leaves’ carbon (C), nitrogen (N), phosphorus (P) content and their stoichiometric ratios in plant association with AM, AM + ECM and ECM under different climate zones.

Biomes	Mycorrhiza Types	C (mg∙g^−1^)	N (mg∙g^−1^)	P (mg∙g^−1^)	N:P Ratio	C:N Ratio	C:P Ratio
Boreal	AM	460.1 ± 27.2 b	8.8 ± 3.6 b	1.7 ± 0.3 a	17.1 ± 10.4 a	54.1 ± 21.1 a	999.2 ± 742.4 a
	AM + ECM	501.4 ± 24.1 a	11.7 ± 4.5 a	1.4 ± 0.3 a	18.6 ± 14.7 a	40.9 ± 18.3 a	1035.0 ± 365.6 a
	ECM	512.3 ± 18.5 a	7.3 ± 3.5 c	1.4 ± 0.3 a	11.8 ± 6.5 b	50.3 ± 14.8 a	1068.5 ± 554.3 a
Temperate	AM	494.5 ± 53.7 a	10.5 ± 4.2 a	1.3 ± 0.3 a	17.0 ± 6.4 a	54.6 ± 22.9 a	702.1 ± 341.3 a
	AM + ECM	498.5 ± 47.6 a	9.8 ± 3.5 a	1.5 ± 0.7 a	16.8 ± 6.8 a	59.8 ± 22.4 a	1216.0 ± 573.6 a
	ECM	503.0 ± 37.0 a	9.5 ± 3.5 a	1.4 ± 0.9 a	17.6 ± 9.4 a	52.8 ± 23.4 a	947.9 ± 795.4 a
Tropical	AM	420.5 ± 51.4 a	12.2 ± 4.2 a	-	28.1 ± 16.4 a	39.7 ± 9.7 b	1325.0 ± 842.9 a
	AM + ECM	466.5 ± 37.1 a	13.3 ± 5.5 a	-	33.4 ± 11.1 a	53.0 ± 5.4 a	1780.0 ± 739.8 a
	ECM	439.0 ± 46.7 a	12.7 ± 3.8 a	-	28.5 ± 17.7 a	35.3 ± 8.5 b	2016.5 ± 2260.6 a

Mean ± SD are reported. Differences between mycorrhizal types were tested using permutation tests of significance. Significant differences (*p* < 0.01) are indicated by different letters.

**Table 2 jof-09-00588-t002:** The senesced leaves’ carbon (C), nitrogen (N), phosphorus (P) content and their stoichiometric ratios in plant associations with AM, AM + ECM and ECM under different plant functional types.

Plant Functional Types	Mycorrhiza Types	C (mg∙g^−1^)	N (mg∙g^−1^)	P (mg∙g^−1^)	N:P Ratio	C:N Ratio	C:P Ratio
Deciduous	AM	434.7 ± 58.3 b	11.5 ± 4.2 a	1.6 ± 0.4 a	16.6 ± 8.4 a	43.3 ± 18.3 a	817.6 ± 610.7 a
	AM + ECM	499.2 ± 37.5 a	10.7 ± 4.5 a	1.5 ± 0.5 a	16.9 ± 11.1 a	48.6 ± 15.8 a	1165.8 ± 468.9 a
	ECM	493.1 ± 36.9 a	10.4 ± 2.9 a	1.4 ± 0.4 a	14.8 ± 10.1 a	49.6 ± 14.7 a	772.9 ± 533.5 a
Evergreen	AM	456.0 ± 68.3 b	10.8 ± 4.2 a	1.2 ± 0.2 a	25.4 ± 16.5 a	48.8 ± 17.6 a	1534.6 ± 877.6 a
	AM + ECM	490.1 ± 46.3 a	11.1 ± 4.2 a	1.3 ± 0.2 a	23.6 ± 11.6 a	58.9 ± 26.0 a	1427.9 ± 657.1 a
	ECM	512.0 ± 39.2 a	7.8 ± 3.9 b	1.2 ± 0.3 a	17.2 ± 9.9 b	56.9 ± 32.5 a	1555.7 ± 1043.6 a
Broadleaf	AM	437.8 ± 59.2 b	11.3 ± 4.2 a	1.5 ± 0.4 a	21.2 ± 13.3 a	44.6 ± 16.5 a	1190.4 ± 830.5 ab
	AM + ECM	493.3 ± 46.1 a	10.8 ± 4.4 a	1.5 ± 0.5 a	21.2 ± 12.0 a	53.6 ± 17.9 a	1459.1 ± 556.2 a
	ECM	492.8 ± 39.9 a	10.4 ± 3.0 a	1.4 ± 0.4 a	17.5 ± 11.2 a	51.2 ± 20.8 a	780.6 ± 523.4 b
Conifer	AM	535.3 ± 53.1 a	10.3 ± 4.0 a	1.4 ± 0.1 a	20.1 ± 17.5 a	60.7 ± 26.8 a	-
	AM + ECM	497.3 ± 29.7 a	11.5 ± 4.3 a	1.3 ± 0.2 a	16.4 ± 10.2 a	54.2 ± 33.8 a	956.8 ± 532.9
	ECM	518.6 ± 30.7 a	7.2 ± 3.7 b	1.3 ± 0.3 a	13.8 ± 7.2 a	55.9 ± 30.0 a	1733 ± 1080.8

Mean ± SD are reported. Differences between mycorrhizal types were tested using permutation tests of significance. Significant differences (*p* < 0.01) are indicated by different letters.

**Table 3 jof-09-00588-t003:** The effect of mycorrhizal type, leaf shape, and their interaction on the element content and their stoichiometric ratios in senesced leaves based on linear mixed effect models.

Leaf Index	Mycorrhizal Type		Leaf Shape		Mycorrhizal Type × Leaf Shape	
	F	*p*	F	*p*	F	*p*
C	35.86	<0.001	7.58	0.03	6.70	0.001
N	15.74	<0.001	13.62	0.03	10.79	<0.001
P	2.50	0.09	0.58	0.45	0.05	0.96
N:P	12.51	<0.001	32.62	<0.001	0.52	0.60
C:N	0.36	0.70	0.55	0.48	0.33	0.72
C:P	0.11	0.90	0.04	0.84	5.92	0.02

Mycorrhizal type, leaf shape and their interaction were fixed effects. Leaf habit was a random effect.

**Table 4 jof-09-00588-t004:** The slope of senesced leaves’ carbon (C), nitrogen (N), phosphorus (P) and their stoichiometric ratios with the increase in MAT among different mycorrhizal types.

Mycorrhizal Types	C and MAT	N and MAT	P and MAT	N:P and MAT	C:N and MAT	C:P and MAT
AM	−4.97 × 10^−3^ b	2.39 × 10^−2^ b	−4.8 × 10^−2^ b	0.88 c	−1.01 b	18.46 b
AM + ECM	−4.1 × 10^−3^ a	1.32 × 10^−2^ c	7.87 × 10^−3^ a	1.14 a	0.73 a	49.85 b
ECM	−1.51 × 10^−3^ a	4.20 × 10^−2^ a	−7.80 × 10^−3^ a	1.02 b	−1.02 a	173.72 a

Numbers provide the slopes. Differences between mycorrhizal types were tested using permutation tests of significance. Significant differences (*p* < 0.01) are indicated by different letters.

**Table 5 jof-09-00588-t005:** The slope of senesced leaves’ carbon (C), nitrogen (N), phosphorus (P) and their stoichiometric ratios with the increase in MAP among different mycorrhizal types.

Mycorrhizal Types	C and MAP	N and MAP	P and MAP	N:P and MAP	C:N and MAP	C:P and MAP
AM	2.49 × 10^−6^ a	1.82 × 10^−4^ c	−1.89 × 10^−4^ b	9.4 × 10^−3^ a	−1.16 × 10^−2^ b	0.45 c
AM + ECM	−4.57 × 10^−5^ c	1.32 × 10^−2^ a	2.97 × 10^−4^ a	8.25 × 10^−3^ b	−5.18 × 10^−3^ a	0.58 b
ECM	−4.73 × 10^−5^ b	3.06 × 10^−4^ b	−2.03 × 10^−4^ b	4.08 × 10^−3^ c	−1.46 × 10^−2^ a	0.77 a

Numbers provide the slopes. Differences between mycorrhizal types were tested using permutation tests of significance. Significant differences (*p* < 0.01) are indicated by different letters.

## Data Availability

All the data supporting the findings of this study are included in this article.

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
