# Peer review of "Influence of Mycorrhiza on C:N:P Stoichiometry in Senesced Leaves"

_jof, 2023, doi:10.3390/jof9050588_

Round 1

Reviewer 1 Report (Previous Reviewer 2)

Below are my comments on the manuscript:

1. I didn't find anything about global trends. how was it tested? Are these words needed in the title?

2. Abstract. Global trends seem to me to be an over-interpretation. Did the authors study mycorrhizae from all over the world? maybe instead of using the word global, write a meta-analysis

3. Keywords should be different than in the title. This increases the search for publications

4. Line 154 Where were the sightings from? Was it a meta-analysis?

5. How was the database created? This methodology is incomprehensible to me. The information must be developed in such a way that it is reproducible by other researchers

6. Unfortunately, the lack of a properly described method makes it difficult to refer to the results and discuss them. That doesn't mean they're bad.

Author Response

Point 1: I didn't find anything about global trends. how was it tested? Are these words needed in the title?

Response 1: Thanks very much for your valuable comments on our manuscript. We have combined a dataset from Yuan and Chen (2009) which include 638 plant species from 112 families at 365 sites and describe the world-wide trends in senesced leaves. Based on the datasets of Yuan and Chen (2009) we classified mycorrhizal types of all the plant species according to the method emplyed by Shi et al. (2020) and Yang et al. (2021). Finally, we established a new dataset, including C, N, P content and their stoichiometric ratio in senesced leaves in forbs, grasses and woody plants with the information on mycorrhizal associations, leaf habits, leaf shapes and biomes. We have revised the title in our manuscript. The title of “Global trends of senesced leaves C: N: P stoichiometry changed with mycorrhizal types” were changed as “The C: N: P stoichiometry in senesced leaves changed with mycorrhizal types”.

Point 2: Abstract. Global trends seem to me to be an over-interpretation. Did the authors study mycorrhizae from all over the world? maybe instead of using the word global, write a meta-analysis.

Response 2: Accepted. Special Thanks to you for your good comments. In our manuscript, the senesced leaves C, N, P content and their stoichiometric ratio data were obtained from the global database established by Yuan and Chen (2009). A new database was established by re-mining Yuan and Chen ‘s database which contain C, N, P content and their stoichiometric ratio in senesced leaves in forbs, grasses and woody plants with the information on mycorrhizal associations, leaf habits, leaf shapes and biomes. We are sorry for our unclear inllustration. You are correct. The words “global trends” may not appropriate in our manuscript. We have deleted the word “global” in Abstract. The specific contents are as follows:

Line 14, The carbon (C), nitrogen (N) and phosphorus (P) stoichiometry in senesced leaves have been reported, which are influenced by biotic and abiotic factors, such as climate variables and plant functional groups. It is well known that mycorrhizal types are one of the most important functional characteristics of plant that affect leaf C: N: P stoichiometry.

Line 19, Here, the patterns in senesced leaves C: N: P stoichiometry among plants associates to arbuscular mycorrhizal (AM), ectomycorrhizal (ECM) or AM+ECM fungi were explored.

Point 3: Keywords should be different than in the title. This increases the search for publications.

Response 3: Accepted. Thank you for your constructive comments. We have revised the keywords to increase the search for publications. The key words of “senesced leaves; mycorrhizal types; stoichiometry; temperature; precipitation” were changed as “biogeochemical cycling, carbon, nitrogen, phosphorus, senesced leaves, mycorrhizal strategy, temperature and precipitation”.

Point 4: Line 154 Where were the sightings from? Was it a meta-analysis?

Response 4: We are sorry for our unclear statement. The data of senesced leaves C, N, P content their stoichiometric ratio we used in our manuscript were obtained from the database eatsblished by Yuan and Chen (2009). We combined a new database which include C, N, P content and their stoichiometric ratio in senesced leaves in forbs, grasses and woody plants with the information on mycorrhizal associations, leaf habits, leaf shapes and biomes based on Yuan and Chen’s database. We have revised the methodology in Materials and Methods. The specific contents are as follows:

Line 116, In this study, the plant senesced leaves C: N: P stoichiometry data were obtained from the global database established by Yuan and Chen [64]. The 1253 observations were obtain from two leaf habits (deciduous vs. evergreen), two leaf shapes (broadleaf vs. conifer leaf shape), and three biomes (boreal, temperate, and tropical). A new database containing senesced leaves C: N: P stoichiometry and associated traits was established by re-mining the database from Yuan and Chen [64].

Point 5: How was the database created? This methodology is incomprehensible to me. The information must be developed in such a way that it is reproducible by other researchers.

Response 5: Accetped. Thank you for your patient and carefully comments. We are sorry for the unclear description in methodology. Firstly, we extracted the data of plant senesced leaves C, N, P content and their stoichiometric and associated leaf habits, leaf shapes, biomes and climate factors from the database eastablished by Yuan and Chen (2009). Secondly, We classified mycorrhizal type of all the plant species according to the method employed by Shi et al. (2020) and Yang et al. (2021). According to all the data we collected, the species forming AM or ECM account for 91.9%. In addition, AM and ECM fungi are two domain functional types inoculated with most plants on Earth. Therefore, we classified the mycorrhizal type as AM, AM+ECM, and ECM. Finally, we established a new database containing senesced leaves C, N, P and their stoichiometric ratios and other associated traits. We have revised the methodology in Materials and Methods to make it more clear. The specific content are as follows:

Line 115,

  1. Materials and Methods

2.1 Data collection

In this study, the plant senesced leaves C: N: P stoichiometry data were obtained from the global database established by Yuan and Chen [64]. The 1253 observations were obtain from two leaf habits (deciduous vs. evergreen), two leaf shapes (broadleaf vs. conifer leaf shape), and three biomes (boreal, temperate, and tropical). A new database containing senesced leaves C: N: P stoichiometry and associated traits was established by re-mining the database from Yuan and Chen [64].

2.2. Mycorrhizal classification

The mycorrhizal type of all the plant species were classified according to the method employed by Shi et al. [16] and Yang et al. [17]. The mycorrhizal types were ascertained according to the published literatures mainly including Wang and Qiu [65], Akhmetzhanova et al. [66], Hempel et al. [67], and Soudzilovskaia et al. [68]. According to all the data we collected, the species forming AM or ECM account for 91.9%. In addition, AM and ECM fungi are two domain functional types inoculated with most plants on Earth [57]. Therefore, we classified the mycorrhizal type as AM, AM+ECM, and ECM. Our data extend to all species to avoid the effect of plant species-specific, ecological, and evolutionary strategies, from the same species in different site to the same species in same site. On this basis, a database was established by combining the identified mycorrhizal types of plant species with data on senesced leaves C, N, P and their stoichiometric ratios and associated traits (Supplementary File). And the database contains 895 observations of mycorrhizal association, deciduous vs. evergreen leaf habit, broadleaf vs. conifer leaf shape, biomes and senesced leaves element content across 397 plant species (Figure 1). In the new senesced leaves dataset, with 244 (61.5%), 45 (11.3%), and 108 (27.2%) plant formed host-specific associations with AM, AM+ECM, or AM+ECM, respectively. Plant species were subdivided into two subgroups based on biomes and plant functional types. The biome was further divided into three sub-subgroups, i.e. boreal, temperate and tropical. The plant functional types were further divided into two sub-subgroups based on leaf shape (i.e. broadleaf and conifer) and leaf habit (i.e. deciduous and evergreen) (Table S1).

Point 6: Unfortunately, the lack of a properly described method makes it difficult to refer to the results and discuss them. That doesn't mean they're bad.

Response 6: Accepted. Thank you very much. We are really sorry for our unclear statement in Materials and Methods. We have revised the describtion of method in our manuscript. Also, we have revised the Results and Disscussion section to make them more clearly. The specific content of method have described above. The detailed revision were in the revised version which already uploaded.

Reviewer 2 Report (Previous Reviewer 1)

In this paper the authors explore the global patterns of carbon (C), nitrogen (N), and phosphorus (P) stoichiometry in senesced leaves of plants associated with different mycorrhizal types, namely arbuscular mycorrhizal (AM), ectomycorrhizal (ECM), or both (AM+ECM) fungi. The study finds that mycorrhizal types are an important functional characteristic of plants that affect leaf C:N:P stoichiometry. In addition to the changes associated with the mean annual temperature (MAT) and mean annual precipitation (MAP) in ECM or AM+ECM plants. Interestingly, in the paper the authors show some evidence that C:N:P stoichiometry depends on mycorrhizal types, and that this changes mainly depending on the temperature change.

In my opinion, the text could be made more understandable by reducing some of the excess description that remains. Additionally, it would be beneficial to discuss the implications of the identified trend in relation to climate change and how these fungi could contribute to the soil's carbon sequestration capacity.

Furthermore, in terms to make more interest the paper for the readers, the discussion would benefit from a more structured approach instead of simply repeating the already presented results. The authors should explore the possible implications of these fungal profiles in the biogeochemical cycle of the three quantified elements and speculate more on their significance.

Author Response

Point 1: In this paper the authors explore the global patterns of carbon (C), nitrogen (N), and phosphorus (P) stoichiometry in senesced leaves of plants associated with different mycorrhizal types, namely arbuscular mycorrhizal (AM), ectomycorrhizal (ECM), or both (AM+ECM) fungi. The study finds that mycorrhizal types are an important functional characteristic of plants that affect leaf C:N:P stoichiometry. In addition to the changes associated with the mean annual temperature (MAT) and mean annual precipitation (MAP) in ECM or AM+ECM plants. Interestingly, in the paper the authors show some evidence that C:N:P stoichiometry depends on mycorrhizal types, and that this changes mainly depending on the temperature change.

Response 1: Thank you for your affirmation of our research. Our research shows that mycorrhizal types alter plant senesced leaves C:N:P stoichiometry and the effect of mycorrhizal types on plant senesced leaves C:N:P stoichiometry is closely related to plant functional groups and climate variables. Our results suggest that senesced leaves C: N: P stoichiometry depends on mycorrhizal types and support the hypothesis that mycorrhizal type is linked to the evolution of plant nutrient economic strategies. Meanwhile, our findings were able to accurately predict that the responses of plant C: N: P stoichiometry in senesced leaves to climate changes among mycorrhizal types. We have made point-to-point revisions to your comments. We have also made revisions in detail at the corresponding position in our manuscript.

Point 2: In my opinion, the text could be made more understandable by reducing some of the excess description that remains. Additionally, it would be beneficial to discuss the implications of the identified trend in relation to climate change and how these fungi could contribute to the soil's carbon sequestration capacity.

Response 2: Accepted. Thanks very much for your valuable comments on our manuscript. We have revised the Results section and Disscustion section by reducing the excess description. The specific content were in the revised version which already uploaded. You are correct. Climate change is altering the interactions among plants and soil organism in ways that will alter the structure and function of ecosystem. It is necessary to discuss the senesced leaves nutrient in relation to climate change among plant association with different mycorrhizal types. Also, the interaction between plants and mycorrhizal fungi respresents a major link between atmospheric and soil cabon capacity. Therefore, we have supplemented the relationship among climate change, mycorrhizal types, and soil carbon in Discussion section. The specific contents are as follows:

Line 372, Moreover, ECM could absorb N from organic matter, while AMF promote microbial de-composition of organic matter to get mineral N [77]. Therefore, the ecosystem domain by ECM had greater soil carbon storage than those do-main by AM [78]. Further, N content in AM+ECM plants were the highest among AM, AM+ECM and ECM plants, indicating that the co-existence of AM and ECM could enhance N absorption by plants.

Line 415, Furthermore, AM plants could promote greater soil carbon loss to improve N uptake from soil than ECM because they enhance organic matter decomposition [88].

Line 432, These results indicated that plant association with different mycorrhizal types had re-markable variability in nutrient acquirement, which also effects on soil C [91].

Line 467, The climate, especially temperature, was the main factor to shape the distribution of AM and ECM host plants [97].

Line 481, Therefore, different mycorrhizal type have different influence on plant adaption to climate change. These results could imply that plants association with different mycorrhizal types contributed differently to nutrient strategies in ecosystem [102].

Point 3: Furthermore, in terms to make more interest the paper for the readers, the discussion would benefit from a more structured approach instead of simply repeating the already presented results. The authors should explore the possible implications of these fungal profiles in the biogeochemical cycle of the three quantified elements and speculate more on their significance.

Response 3: Accepted. It is realy ture as Reviewer suggested that the discussion should form a structured approach. Also, our research is consistent with an eco-evolutionary feedback between plants and the soil nutrient environment, mediated by mycorrhizal fungi, selecting for plant traits that alter nutrient cycling within AM and EM ecosystems. So, it is important to explore that the significance of mycorrhizal types on explain the dynamics of biogeochemical cycles. We have revised the Disscussion section in our manuscript. The specific contents are as follows:

Line 30, Our results suggest that senesced leaves C: N: P stoichiometry depends on mycorrhizal types and support the hypothesis that mycorrhizal type is linked to the evolution of carbon-nutrient cycle interactions in ecosystem.

Line 335, Our results showed that mycorrhizal types affected senesced leaves C ,and N, and N: P ra-tio. According to Averill et al.’s research [37], the N content in senesced leaves also presented significant difference between AM and ECM plants. These similar results indicated that patterns of nutrient cycles which related to mycorrhizal types may be ecosystem-specific [71].

Line 401, These findings imply that the mycorrhizal symbiosis could represent carbon-nutrient cycle interactions in ecosystem because that the plant association with different mycorrhizal types had different nutrient acquisition strategies [86-87].

Line 461, This result shows that mycorrhizal types present a larger control over senesced leaves C and N content than leaf shape. This resultIt may be caused by the different effect of mycorrhiza on plant resorption.

Line 484, Considering the distinct patterns of climate factors and senesced leaves nutrient among plants association with different mycorrhizal types, our findings suggested that mycorrhizal strategies can explain the pathways of plant nutrient acquisition among species under climate change.

Besides, we also supplemented some references related to the effect on biogeochemical cycle of three mycorrhizal types.

  1. Xu, J.-W., Lin, G., Liu, B., Mao, R. Linking leaf nutrient resorption and litter decomposition to plant mycorrhizal associ-ations in boreal peatlands. Plant and Soil. 2020, 448, 413-424, doi:10.1007/s11104-020-04449-9.
  2. Averill, C., and C. V. Hawkes. Ectomycorrhizal fungi slow soil carbon cycling. Ecol Lett. 2016, 19, 937–947, doi:10.1111/ele.12631.
  3. Orwin, K. H., Kirschbaum, M. U. F., St John, M. G. & Dickie, I. A. Organic nutrient uptake by mycorrhizal fungi en-hances ecosystem carbon storage: a model-based assessment. Ecol. Lett. 2011, 14, 493–502, doi:10.1111/j.1461-0248.2011.01611.x.
  4. M. Shi, J. B. Fisher, E. R. Brzostek, R. P. Phillips. Carbon cost of plant nitrogen acquisition: Global carbon cycle impact from an improved plant nitrogen cycle in the Community Land Model. Glob. Change Biol. 2016, 22, 1299–1314, doi:10.1111/gcb.13131.
  5. Terrer, C., Vicca, S., Stocker, B.D., Hungate, B.A., Phillips, R.P., Reich, P.B., Finzi, A.C., Prentice, I.C. Ecosystem respons-es to elevated CO2 governed by plant-soil interactions and the cost of nitrogen acquisition. New Phytol. 2018, 217, 507–522, doi: 10.1111/nph.14872.
  6. Wurzburger, N., Brookshire, E. N. J. Experimental evidence that mycorrhizal nitrogen strategies affect soil carbon. Ecol., 2017, 98, 1491–1497, doi:10.1002/ecy.1827.
  7. Chen, W., R. T. Koide, T. S. Adams, J. L. DeForest, L. Cheng, and D. M. Eissenstat. Root morphology and mycorrhizal symbioses together shape nutrient foraging strategies of temperate trees. PNAS. 2016, 113:8741–8746, doi: 10.1073/pnas.1601006113.
  8. Barceló, M., van Bodegom, P. M., Soudzilovskaia, N. A. Climate drives the spatial distribution of mycorrhizal host plants in terrestrial ecosystems. J. of Ecol. 2019, 107, 2564-2573, doi:10.1111/1365-2745.13275.
  9. Bennett, A. E., Classen, A. T. Climate change influences mycorrhizal fungal–plant interactions, but conclusions are limited by geographical study bias. Ecol. 2020, e02978, doi:10.1002/ecy.2978.

Round 2

Reviewer 1 Report (Previous Reviewer 2)

Thank you for submitting the revised version of the manuscript. I can see improvement, but a few things could still be improved.

1. Title: I propose to start with the influence of mycorrhiza on .....

2. Check that all tables and figures are referenced in the text

Author Response

Thank you for submitting the revised version of the manuscript. I can see improvement, but a few things could still be improved.

  1. Title: I propose to start with the influence of mycorrhiza on .....

Response: Accepted. We agree to change our title into “Influence of mycorrhiza on C:N:P stoichiometry in senesced leaves”.

  1. Check that all tables and figures are referenced in the text.

Response: Thank you for your reminder and great help. We have checked all tables and figures.

This manuscript is a resubmission of an earlier submission. The following is a list of the peer review reports and author responses from that submission.

Round 1

Reviewer 1 Report

In this work, the authors show the stoichiometry of C:N:P in senescent leaves in plants. Also, what is the effect of the symbiosis with different types of mycorrhizal fungi and the effect of the biome. But, although the data are interesting, the way the data is written and presented in the manuscript can be quite confusing. That is why I recommend presenting the most relevant data to lead to the general conclusions of the work.

Some tables may be sent to supplementary material and later only refer to the observed results.

Reviewer 2 Report

The work sent for review should be rewritten. There is nothing in the introduction about which plants the material was taken from. Mycorrhizal species will have a different effect on plants, i.e. grasses, corn or trees. Only in the methodology do we learn that these are evergreen plants. Saprotrophs and endotrophic were omitted in the correlation of the element content with mycorrhizal fungi. These groups will also affect the element content in the leaves. The results are exciting and well presented graphically, but to improve readability, they should be reworded manuscript.